# Soil quality enhancement by multi-treatment in the abandoned land of dry-hot river valley hydropower station construction area under karst desertification environment

**Qinglin Wu[1], Rong Sun[2], Fan Chen[2], Xichuan Zhang[2], Panpan Wu[1], Lan Wang[1,3], Rui Li[1] ***

1 School of Karst Science, Guizhou Normal University/State Engineering Technology Institute for Karst Desertification Control, Guiyang, China, 2 Power China Guiyang Engineering Corporation Limited, Guiyang, China, 3 School of Foreign Languages, Guizhou Normal University, Guiyang, China

* rlfer77@163.com

**Data Availability Statement:** All relevant data are within the paper and its Supporting Information files.

## Abstract

The medium-intensity karst desertification environment is typically characterized by more rocks and less soil. The abandoned land in the construction areas of the dry-hot river valley hydropower station has more infertile soil, severe land degradation, and very low land productivity. Therefore, it is urgent to improve the soil quality to curb the increasingly degrading land and reuse the construction site. Few studies have focused on the effect of soil restoration and comprehensive evaluation of soil quality with multi-treatment in abandoned land in the dry-hot valley hydropower station construction area. Here, 9 soil restoration measures and 1 control group were installed at the Guangzhao Hydropower Station construction in Guizhou Province, China, for physical and chemical property analysis. In total, 180 physical and 90 chemical soil samples were collected on three occasions in May, August, and December 2022. Soil fertility and quality were evaluated under various measures using membership functions and principal component analysis (PCA). This study showed that almost all measures could enhance soil water storage capacity (The average total soil porosity of 9 soil treatments was 57.56%, while that of the control group was 56.37%). With the increase in soil porosity, soil evaporation became stronger, and soil water content decreased. Nevertheless, no decrease in soil water content was observed in the presence of vegetation cover (soil water content: 16.46% of hairy vetch, 13.99% of clover, 13.77% of the control). They also proved that manure, synthetic fertilizer, and straw could promote total and available nutrients (Soil total nutrient content, or the total content of TN、TP、TK, was presented as: synthetic fertilizer (11.039g kg$^{-2}$)>fowl manure (10.953g kg$^{-2}$)>maize straw (10.560g kg$^{-2}$)>control (9.580g kg$^{-2}$);Total available nutrient content in soil, or the total content of AN,AP,A,was shown as:fowl manure (1287.670 mg kg$^{-1}$)>synthetic fertilizer (925.889 mg kg$^{-1}$)>sheep manure (825.979 mg kg$^{-1}$)>control (445.486 mg kg$^{-1}$). They could also promote soil fertility, among which the first two reached the higher comprehensive soil quality. Fertilizer was conducive to improve soil quality and fertility, yet long-term application could cause land degradation like soil non-point source pollution, compaction, and land

**Funding:** This study was supported by the Technology Support Plan of Guizhou Province (No. 462 2021 Qiankehe Zhicheng) and the Natural Science Foundation of Guizhou Province (No. 317 2022 Qiankehe Jichu -ZK). The funders had no role in study design, data collection and analysis, decision to publish, or preparation of the manuscript.

**Competing interests:** The authors have declared that no competing interests exist.

productivity decline. Ultimately, combining fertilizer with biochar or manure is recommended to improve soil fertility. Biochar and green manure could play an apparent role in soil improvement only when there is abundant soil water. The above views provide theoretical support for curbing soil degradation, improving soil fertility and quality, enhancing land productivity, and promoting the virtuous cycle of the soil ecosystem.

# 1 Introduction

The global karst area is about 22 million square kilometres, accounting for 15% of the land area [1, 2] and providing water resources for about 20–25% of the global population [3]. In China, Karst is mainly distributed in its southwest regions. It has the widest area of exposed carbonate rocks globally, with 220 million people living here [4, 5]. Karst is one of the most fragile terrestrial ecosystems in the world [6]. Karst desertification is the fundamental cause of ecological fragility in karst areas. It is also an extreme manifestation of land degradation [7], the result of environmental degradation formed under unique hydrological and geological conditions, and a phenomenon of land degradation resulting from natural factors such as rainfall, soil erosion and human activities [2]. Karst areas are characterized by high permeability, which makes it difficult to collect water on the surface and causes water resources to easily leak into the ground through surface cracks [8]. This distribution characteristic of water resources leads to a decrease in surface water, resulting in abundant groundwater that is difficult to use, presenting a unique 'karst drought' [6]. Typically, a dry and hot river valley is a valley with high temperature and low humidity. It features sufficient light and heat resources, high temperature and drought climate, low annual rainfall but extensive evaporation, prominent water-heat contradiction, low vegetation coverage rate, severe soil erosion, ecological degradation, and difficult vegetation restoration [9]. The soil layer (about 20 cm) is skinny in the hydropower station construction area in the karst desertification environment. The abandoned land in the dry-hot valley is mixed with gravel and broken bricks left over from the hydropower station construction. This results in severe land degradation, thereby decreasing the soil organic matter due to the gravel (In general, the increase in gravel caused a decrease in soil water content, coverage, above-ground and underground biomass, community density, and species richness, even leading to the extinction of some species.) [10], which further leads to even more infertile soil with almost no land productivity. Evidently, soil is an indispensable resource for human survival and is critical in providing nutrients for plant growth and supporting human well-being [11]. Its degradation seriously hinders the development of ecological and sustainable agriculture in hydropower stations, so it is imperative to improve its quality.

Soil improvement has been intensively investigated, yet most studies have mainly been concerned with the actual effect of different materials and measures. A study reported that the biochar-compost practice reduced the high mineralization rate of soil organic matter, phosphorus deficiency and aluminium toxicity, improving soil's physical and chemical properties [12]. It was also claimed in other research that biochar can improve the soil microbial community structure and metabolites [13, 14] and promote soil cation exchange capacity to reach higher nutrient availability and stronger soil nutrient retention ability [15]. Winter rye is also helpful in soil restoration, and its long-term mulching helps accumulate more soil moisture [16]. Besides, a report claims that soil could be improved by sludge from sewage treatment plants, which can reduce the void index and improve particle packaging of soil [17]. Organic polymer soil conditioners are another commonly used measure of soil improvement. A synthetic

consortium was reported to be beneficial in agricultural soil due to its ability to increase soil fertility, reduce soil pollution, and promote soil productivity [18]. Water gel releases nitrogen, phosphorus, and potassium into the soil over an extended period, facilitating plant growth, enhancing nutrient levels, and inhibiting the accumulation of harmful nitrates in plants [19]. Plant urease was found to induce calcium carbonate precipitation to achieve soil improvement [20]. Polyacrylamide, sodium polyacrylate, hydroxypropyl methylcellulose and polyvinyl alcohol were applied as raw materials due to being able to lower soil erosion rate and increase plant biomass and soil porosity [21].

The prompt initiation of soil remediation is imperative when soil degradation is present due to human activities within a particular environment, such as soil damage caused by mining or the construction of hydropower stations. Some previous studies have focused on mining-damaged soil and its restoration [22, 23]. It was found that the contents of available nitrogen (AN), available phosphorus (AP) and available potassium (AK) in coal mining areas were significantly lower than those in non-mining ones, which indicated that mining activities destroyed the soil architecture, resulting in weakened soil water holding capacity, more evaporation and lower soil water content [24]. Another study showed that coal mining contaminated soil could be improved by applying a microorganism activation method to achieve a more extensive content of AP, AK and available silicon (ASi) [25]. Also, it was reported that municipal sludge could be used for soil restoration on the planting sites of the Alquife Mine for tomatoes (*Lycopersicon esculentum Mill.*), ryegrass (*Lolium perenne L.*), and ahipas (*Pachyrhizus ahipa.*) (*Wedd.*) *Parodi*), producing various effects with different metering sludge composting [26]. Currently, soil improvement on hydropower station sites has received limited attention. Meanwhile, the current studies on soil improvement are mainly based on researching and developing new laboratory materials, which are challenging to promote and apply in agricultural production [27]. Obviously, previous studies rarely took multiple evaluation methods to comprehensively evaluate soil fertility and soil quality using multiple soil improvement treatments. Therefore, this study took the abandoned land on the hydropower station construction area in a karst desertification environment under a dry-hot river valley landform type as the research object. In addition, traditional manure, green manure, synthetic fertilizers, biochar and straw were selected to conduct field experiments on soil remediation. This study employs data-analysis methods, including membership function, soil quality index, and soil nutrient level classification. The objective of this study is to comprehensively evaluate soil fertility and quality, aiming to derive practical soil improvement measures tailored to specific climate and geomorphic types. This study seeks to offer theoretical support for the enhancement of degraded land, the establishment of a virtuous cycle in the soil ecosystem, the control of karst desertification, and the development of ecological agriculture.

## 2 Materials and methods

### 2.1 Overview of the study area

Guangzhao Hydropower Station (105˚15 '00 "E, 25˚37' 34" N) is located in the midstream of the Beipan River at the junction of Guanling and Qinglong County, Guizhou Province, China, 222 km away from Guiyang (Fig 1), with 13,548 km$^2$ rainfall collection area of the dam site and an average annual rainfall of 1178.8 mm. It is the largest hydropower station of cascade development projects in the Beipan River mainstream and the critical project in the second batch of Chinese 'west-east power transmission' construction. The dam site is a karst mountain terrain, with an elevation difference of over 1000 m from the plateau top to the valley bottom. It is a typical V-shaped river valley with Triassic strata and limestone as the primary lithology, belonging to medium-intensity level karst desertification environment. The land use

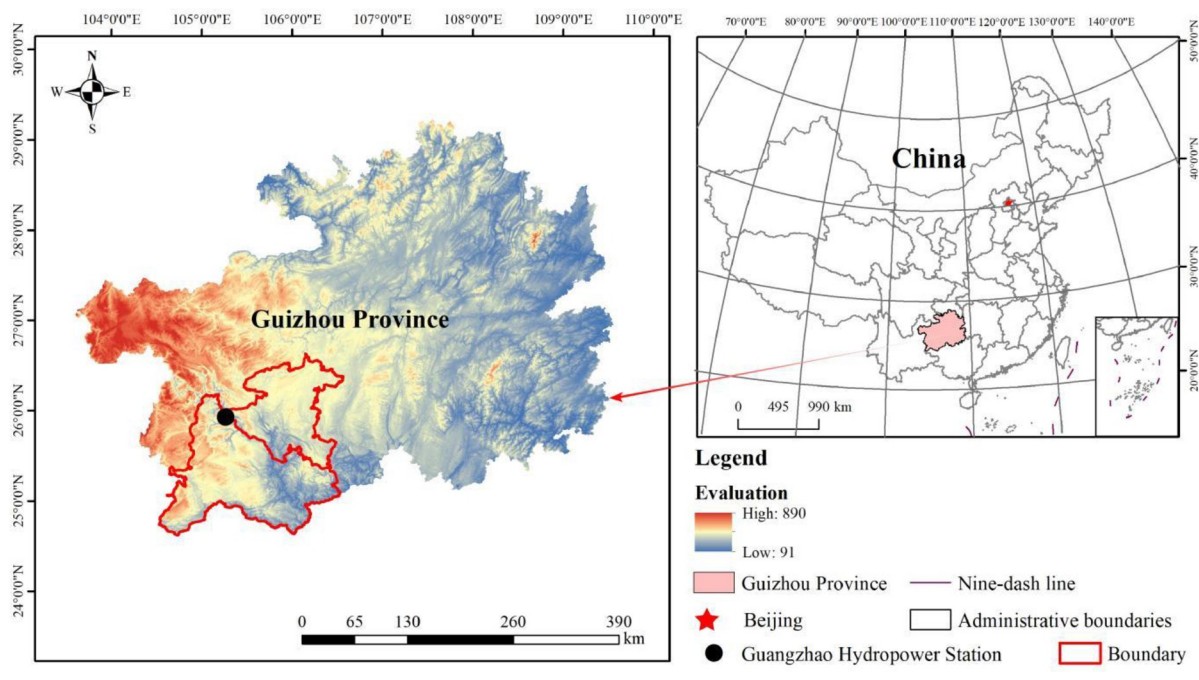

**Fig 1. Location map of the research area.**

types here include grassland, water area, forest, cultivated land, construction land and bare rock. The poor vegetation in the upstream watershed of the hydropower station has caused severe soil erosion. The erosion area of the reservoir has reached 2844.72 ha, with a construction waste area of 181.83 ha.

## 2.2 Experimental design

In December 2021, 10 strip plots (1 m × 4 m) with a 5° slope were selected to arrange soil improvement measures; where nine of them were tested treatments, and another one was a control plot (Table 1 and Fig 2). The nine treatments were selected based on the following reasons. First, hairy vetch and clover were supposed to be common green manure in karst areas and thus chosen as biological measures for soil restoration. Second, the easy availability of maize straw, maize straw biochar, rice husk biochar, fowl manure, sheep manure, and synthetic fertilizer in karst mountainous areas makes them the chosen soil-restoring materials. Therefore, measures were selected according to adapting to local conditions. The soil samples were taken three times in May, August and December 2022. When taking the soil for physical property measurement, they were sampled at three points in each of the ten sample plots (including the control plot) along an S-shaped path with a 100 cm$^3$ ring knife. At each point, the soil was taken from two different depths of the soil layer (0–10 cm and 10–20 cm). In this way, 60 physical soil samples were taken each time, and a total of 180 ones were obtained three times. With the same sampling path and at the same point as physical soil, the sample was also taken from two soil layers (0–10 cm and 10–20 cm) with scraper knives for chemical property measurement. At each point in each layer, 200 g of soil was taken. After that, soil from the three points in the same layer was mixed and put in a ziploc bag. In this way, 30 chemical soil samples were collected each time, and 90 were obtained. After that, they were returned to the laboratory for air drying and impurity removal and passed through a 100-mesh sieve for testing.

**Table 1. Basic information of the sample plots.**

| Measures | Concrete treatments |
|---|---|
| Hairy vetch | The sowing rate was 7.5 g m⁻². Before sowing, the soil was ploughed and leveled up. Cover depth was 4–6 cm after seeding. |
| Rice husk biochar | The application rate was 1 kg m⁻², mixed well with 10 cm of topsoil and then leveled up. |
| Maize straw biochar | The application rate was 1 kg m⁻², mixed well with 10 cm of topsoil and then leveled up. |
| Maize straw | The application rate was 1.2 kg m⁻², mixed well with 10 cm of topsoil and then leveled up. |
| Maize straw + Hairy vetch | The application rate of maize straw was 0.6 kg m⁻², mixed well with 10 cm of topsoil and then leveled up. |
| | The sowing rate of hairy vetch was 38 g m⁻². Before sowing, the soil was ploughed and leveled up. The cover depth was 10 cm after seeding. |
| Fowl manure | Manure was applied according to the Technical Specification for Returning Livestock and Poultry Manure to Field (GB-T 25246–2010), and the application rate was 1.67 kg m⁻². |
| Sheep manure | Sheep manure was applied according to the technical specification for returning livestock and poultry manure to the field (GB-T 25246–2010), and the application rate was 1.67 kg m⁻². |
| Clover | The sowing rate was 7.5 g m⁻², and the soil was ploughed and leveled. Cover depth was 4–6 cm after seeding. |
| Synthetic fertilizers | The N–P2O5–K2O ratio was 15:15:15, and the application rate was 800 kg ha⁻¹, mixed well with 10 cm of topsoil and then leveled up. |
| Control | No treatment |

## 2.3 Sample analysis

The potassium dichromate volumetric method-external heating method was used to determine the SOM. The micro Kjeldahl method was used to test the TN, and the alkaline hydrolysis diffusion method was used to measure the AN. The TP content of the soil was tested using ultraviolet spectrophotometer, and the AP content was tested using the molybdenum-antimonic resistance colorimetric method. Flame spectrophotometry was employed to assess the

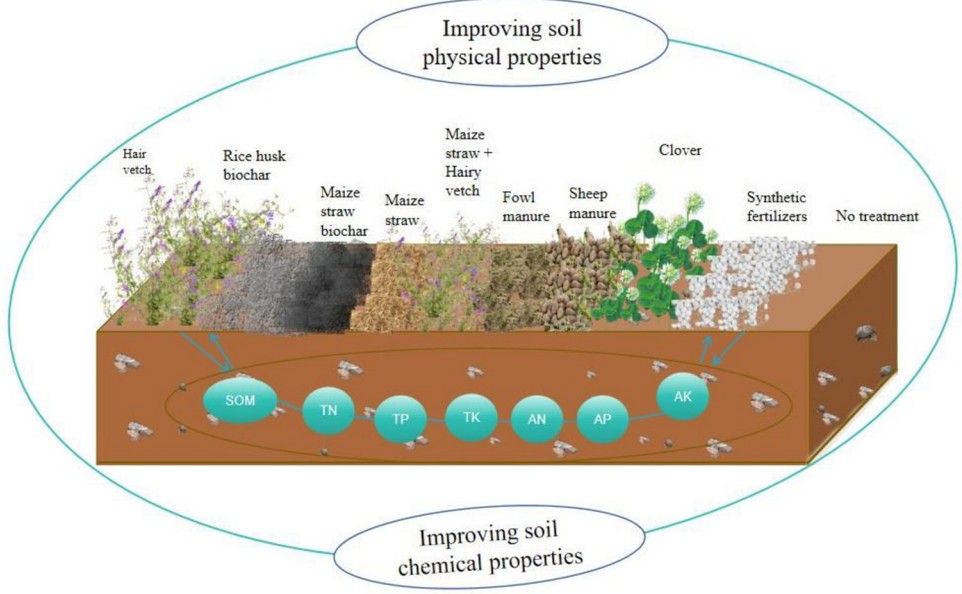

**Fig 2. Soil quality enhancement mechanism map.**

content of TK and AK, and the potentiometric method (1:5) was used to determine the pH value of soil [28, 29].

## 2.4 Data analysis

**2.4.1 Classification standard of soil nutrient index.** According to the nutrient classification standard of the second National Soil Survey (National Earth System Science Data Center), the soil nutrient level in the study areas was evaluated (Table 2) [30].

**2.4.2 Selection and data calculation of soil comprehensive evaluation indexes.** The physical indexes considered for the comprehensive soil evaluation included soil water content, saturated water content, capillary water content, field water capacity, bulk density, total porosity and capillary porosity. The selected chemical indexes were SOM, TN, TP, TK, AN, AP, AK, and pH. In total, 15 physical and chemical indexes were evaluated by the soil quality index formula (Formula 1) [31, 32].

$$SQI = \sum_{i=1}^{n} S_i W_i \tag{1}$$

Where $SQI$ is the soil quality index, $S_i$ is the membership function value of the $i_{th}$ index, and $W_i$ is the weight of the $i_{th}$ index. The higher the $SQI$, the better the soil quality [31, 32]. The PCA obtained the index weights using IBM SPSS Statistics 23 [33]. Since the index change was continuous in soil quality evaluation, the membership function was used to evaluate it. When the soil indexes had a positive effect on soil productivity, the ascending distribution function was applied to the calculation (Formula 2) [34, 35]:

$$f(x_i) = \begin{cases} 1.0 & x_i \geq x_2 \\ 0.1 + 0.9(x_i - x_1)/(x_2 - x_1) & x_1 < x_i < x_2 \\ 0.1 & x_i \leq x_1 \end{cases} \tag{2}$$

When the soil indexes had adverse effects on soil productivity, the descending distribution function was selected for the calculation, as shown in equation (Formula 3) [34, 35]:

$$f(x_i) = \begin{cases} 0.1 & x_i \geq x_2 \\ 0.1 + 0.9(x_2 - x_i)/(x_2 - x_1) & x_1 < x_i < x_2 \\ 1.0 & x_i \leq x_1 \end{cases} \tag{3}$$

Where $f(x_i)$ represents the membership degree value of the index, $x$ for the actual value of each index, $x_1$ and $x_2$ for the minimum and maximum values in the index, respectively, and $x_i$ for the measured value. The membership degrees of soil physical attributes, except for bulk density, were calculated by ascending distribution function, and only the pH value was

**Table 2. Nutrient level of the second national soil survey.**

| Index/level | 1 | 2 | 3 | 4 | 5 | 6 |
|---|---|---|---|---|---|---|
| SOM (g kg⁻¹) | >40 | 30–40 | 20–30 | 10–20 | 6–10 | <6 |
| TN (g kg⁻¹) | >2 | 1.5–2 | 1–1.5 | 0.75–1 | 0.5–0.75 | <0.5 |
| TP (g kg⁻¹) | >2 | 1.5–2 | 1–1.5 | 0.75–1 | 0.5–0.75 | <0.5 |
| TK (g kg⁻¹) | >20 | 15–20 | 10–15 | 5–10 | 3–5 | <3 |
| AN (mg kg⁻¹) | >150 | 120–150 | 90–120 | 60–90 | 30–60 | <30 |
| AP (mg kg⁻¹) | >40 | 20–40 | 10–20 | 5–10 | 3–5 | <3 |
| AK (mg kg⁻¹) | >200 | 150–200 | 100–150 | 50–100 | 30–50 | <30 |

calculated by descending distribution function for the membership degree of chemical attributes [34, 35].

# 3 Results and analysis

## 3.1 Soil water storage capacity

Soil porosity is an important index to evaluate soil water conservation capacity [36] and soil infiltration capacity [37]. It is also a place for water storage and a channel for water migration [38]. Increasing soil porosity will inevitably lead to enhancing soil infiltration capacity and soil water storage capacity [39]. At the studied construction site in Guangzhao hydropower station, the total soil porosity under 9 soil improvement treatments was 57.76% on average (with that of the control being 56.37%), with 64.92% at the highest and 49.75% at the lowest (Fig 3). The 0–10 cm soil layer (57.24%) was lower than the 10–20 cm soil layer (58%), and there was no significant difference in total porosity between them. Although there was no significant correlation between soil water content and total porosity (r = -0.226, P = 0.082), there was still a negative relation between total porosity and soil water content in some soil improvement measures. For instance, the total porosity of maize straw (59.32%) was equal to that of maize straw + hairy vetch (59.32%), both higher than other measures-synthetic fertilizer (58.54%), rice husk biochar (57.04%), and sheep manure (56.37%). In contrast, the soil water contents gradually declined from sheep manure (16.98%), rice husk biochar (16.84%), synthetic fertilizer (14.45%), maize straw (11.98%), to maize straw + hairy leaves (11.35%). The greater the porosity, the more intense the soil water evaporation occurs, leading to a decline in soil water content. However, the situation with vegetation cover was different. With vegetation coverage, the total porosity of soil was ranked from high to low as follows: clover (58.37%), hairy vetch (56.78%), and control (54.78%). Instead, the soil water contents were found as hairy vetch (16.46%) > clover (13.99%) > control (13.77%). This indicates that vegetation cover could not

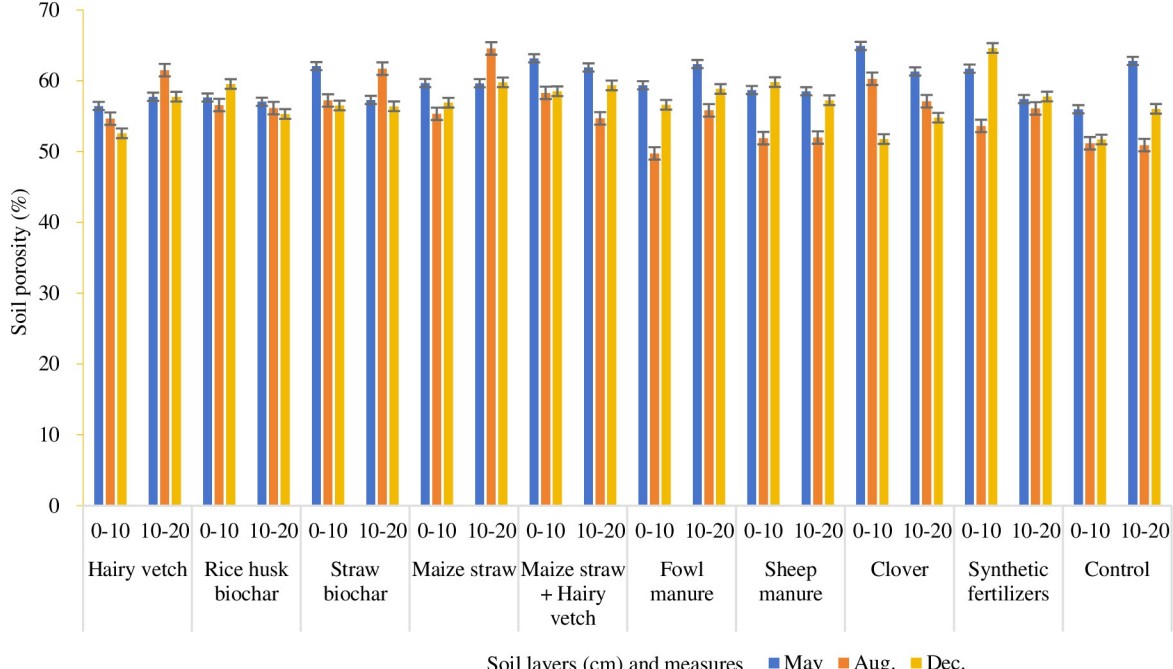

**Fig 3. Soil total porosity under soil improve measures.**

reduce soil water content due to large porosity but could effectively inhibit soil evaporation and retain water better. Compared with the Huajiang karst desertification control demonstration area, which is also a dry-hot river valley under medium-intensity karst desertification environment but with no construction activity, the soil porosity (57.62%) in the construction site was higher than that in non-construction land (soil porosity: 35.63%), whereas the reverse is true for soil water content, with 14.29% in construction site and 17.83% in non-construction site [40].

## 3.2 Soil nutrient characteristics under soil improvement measures

**3.2.1 Soil organic matter.** The ideal soil for plant growth might have a loose structure, sufficient pore space, and relatively rich organic matter [41]. As represented in Table 3, the average content of the SOM was 29.28 g kg⁻¹, with sheep manure (37.30 g kg⁻¹) at the highest and hairy vetch (19.80 g kg⁻¹) at the lowest. Except for hairy vetch, the soil organic matter contents of other treatments were higher than that of control. Hairy vetch had lower content than that of the control, for the possible reason that no ploughing was done before planting, and it increased rapidly, absorbing and carrying away the SOM in the soil. Regarding testing time, the SOM content gradually ascended from May to December under most treatments (rice husk biochar, maize straw biochar, maize straw, maize straw + hairy vetch, sheep manure, and synthetic fertilizer). This shows that the decomposed organic matter of straw, biochar, and sheep manure increased with time within a year. Contrary to the previous measures, the content under the fowl manure measure decreased from May to December, indicating that the SOM had been decomposed mainly in May. In the later period, the content continued to decrease with the consumption of the SOM by weeds and other plants in the land, as well as the washing action and eluviation of the SOM by rainfall. The SOM content in the 0–10 cm

**Table 3. Content of soil organic matter.**

| Measures | Soil layers (cm) | SOM g kg⁻¹ | | | |
|---|---|---|---|---|---|
| | | May | Aug. | Dec. | Average |
| Hairy vetch | 0–10 | 26.92 | 26.25 | 25.67 | 26.28 |
| | 0–20 | 10.48 | 16.66 | 12.84 | 13.32 |
| Rice husk biochar | 0–10 | 33.25 | 29.08 | 39.69 | 34.01 |
| | 0–20 | 11.59 | 21.49 | 24.92 | 19.33 |
| Straw biochar | 0–10 | 36.17 | 31.40 | 36.31 | 34.62 |
| | 0–20 | 19.40 | 24.77 | 28.19 | 24.12 |
| Maize straw | 0–10 | 36.97 | 37.71 | 34.87 | 36.52 |
| | 0–20 | 16.90 | 21.05 | 35.10 | 24.35 |
| Maize straw + Hairy vetch | 0–10 | 39.75 | 47.94 | 38.84 | 42.18 |
| | 0–20 | 23.73 | 18.73 | 40.44 | 27.63 |
| Fowl manure | 0–10 | 49.89 | 39.67 | 32.81 | 40.79 |
| | 0–20 | 20.77 | 21.54 | 28.33 | 23.55 |
| Sheep manure | 0–10 | 19.55 | 52.03 | 56.38 | 42.65 |
| | 0–20 | 19.88 | 48.01 | 27.92 | 31.94 |
| Clover | 0–10 | 43.53 | 33.26 | 36.71 | 37.83 |
| | 0–20 | 21.97 | 26.74 | 25.80 | 24.84 |
| Synthetic fertilizers | 0–10 | 25.19 | 25.81 | 28.56 | 26.52 |
| | 0–20 | 22.55 | 27.68 | 29.64 | 26.62 |
| Control | 0–10 | 33.54 | 24.86 | 27.87 | 28.75 |
| | 0–20 | 14.44 | 26.79 | 17.91 | 19.71 |

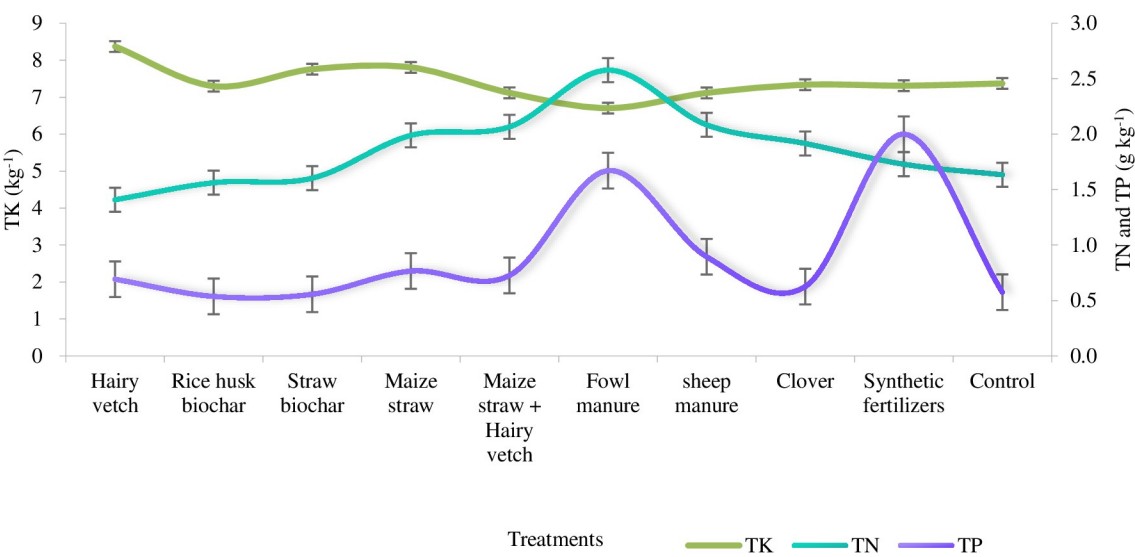

**Fig 4. Soil total nutrients under soil improving measures.**

soil layer was higher than that in the 10–20 cm soil layer, opposite to soil pH distribution. Comparison among different treatments showed that the contents of the SOM were descending in the following order: sheep manure (37.30 g kg-1), maize straw + hairy vetch (34.91 g kg$^{-1}$), fowl manure (32.17 g kg$^{-1}$), clover (31.34 g kg$^{-1}$), maize straw (30.44 g kg$^{-1}$), straw biochar (29.37 g kg$^{-1}$), Rice husk biochar (26.67 g kg$^{-1}$), synthetic fertilizer (26.57 g kg$^{-1}$), control group (24.23 g kg$^{-1}$), and hairy vetch (19.80 g kg$^{-1}$).

**3.2.2 Total soil nutrients.** The average TN content in the soil was 1.857 g kg$^{-2}$, with the highest observed in fowl manure (2.577 g kg$^{-2}$) and the lowest in hairy vetch (1.408 g kg$^{-2}$). Among the nine treatments, the TN contents of maize straw, maize straw + hairy vetch, sheep manure, clover and synthetic fertilizer were higher than that of the control, suggesting that these measures were beneficial to the enhancement of the TN content. Hairy vetch, rice husk biochar and maize straw biochar exhibited lower TN contents than the control. This is due to its strong soil nitrogen absorption ability, leading to the loss of nitrogen from the soil when there is no tillage [42]. For the two biochar measures, the less the TN indicated that the application of biochar brought little benefit to the increase of the TN. The average content of the TP was 0.904 g kg$^{-2}$. The highest was synthetic fertilizer (1.999 g kg$^{-2}$), and the lowest was rice husk biochar (0.537 g kg$^{-2}$). Except for two types of biochar, soil TP levels under alternative measures were higher than those in the control group. This implies that biochar had minimal impact on enhancing TP content, whereas other treatments exhibited notable effects. The average content of the TK in soil was 7.419 g kg$^{-2}$, with hairy vetch showing the highest value (8.370 g kg$^{-2}$) and fowl manure exhibiting the lowest (6.705 g kg$^{-2}$). Hairy vetch, rice husk biochar and maize straw biochar presented a higher TK content than the control (Fig 4). This indicates that biochar was favorable to improving the TK content, and the best was hairy vetch. Overall, the TK (7.419 g kg$^{-2}$) was more than the TN (1.857 g kg$^{-2}$), followed by the TP (0.904 g kg$^{-2}$). Vertically, the contents of TN, TP and TK in the 0–10 cm soil layer (2.199 g kg$^{-2}$, 1.010 g kg$^{-2}$, 7.691 g kg$^{-2}$) were higher than those in the 10–20 cm soil layer (1.515 g kg$^{-2}$, 0.799 g kg$^{-2}$, 7.146g kg$^{-2}$). Regarding the time scale, the highest contents of the TN and TP appeared in August, ascending from May to August and descending from August to December. On the whole, soil total nutrient contents were shown as synthetic fertilizer (11.039g kg$^{-2}$) > fowl manure (10.953 g kg$^{-2}$) > maize straw (10.560 g kg$^{-2}$) > hairy vetch (10.470 g kg$^{-2}$) >

sheep manure (10.094 g kg$^{-2}$) > maize straw biochar (9.916 g kg$^{-2}$) > maize straw + hairy vetch (9.910 g kg$^{-2}$) > clover (9.879 g kg$^{-2}$) > control group (9.580 g kg$^{-2}$) > rice husk biochar (9.399 g kg$^{-2}$).

**3.2.3 Soil available nutrients.**    Fig 5 depicts the contents of soil available nutrients—AK, AN, AP. The highest were 851.872, 276.749, and 159.049 mg Kg$^{-1}$, while the lowest were 207.020, 139.412, and 15.981 mg Kg$^{-1}$, respectively. The AN contents in soil ranged from 139.412 mg kg$^{-1}$ to 276.749 mg kg$^{-1}$, with the AN average value of 190.338 mg kg$^{-1}$. AN content was the highest in soil under fowl manure treatment. Except for hairy vetch, rice husk biochar and maize straw biochar, the contents of AN in other treatments were higher than that of the control (164.197 mg kg$^{-1}$). Vertically, the 0–10 cm soil layer had higher average AN content (225.948 mg kg$^{-1}$) than the 10–20 cm soil layer (154.728 mg kg$^{-1}$). The contents of the AP in soil ranged from 15.981 mg kg$^{-1}$ to 159.049 mg kg$^{-1}$, with an average of 60.980 mg kg$^{-1}$ and the peak in soil treated with fowl manure. Under all treatments, the AP was higher than that of the control, indicating that these soil improvement measures contributed to the increase of soil AP content. Comparison between soil layers showed that the average AP in the upper layer (65.569 mg kg$^{-1}$) was higher than in the lower layer (56.392 mg kg$^{-1}$). The AK in soil ranged from 207.020 mg kg$^{-1}$ to 851.872 mg kg$^{-1}$, with an average of 407.731 mg kg$^{-1}$ and the highest in soil under sheep manure treatment. Some measures (with hairy vetch, rice husk biochar and maize straw biochar excluded) had higher AK than the control. The reason was that hairy vetch with strong absorption ability, rice husk biochar, and maize straw biochar took in part of AK in soil, thus decreasing the AK content [43]. Vertically, the AK was more in the upper layer (516.274 mg kg$^{-1}$) than in the lower layer (299.188 mg kg$^{-1}$). On the whole, the total

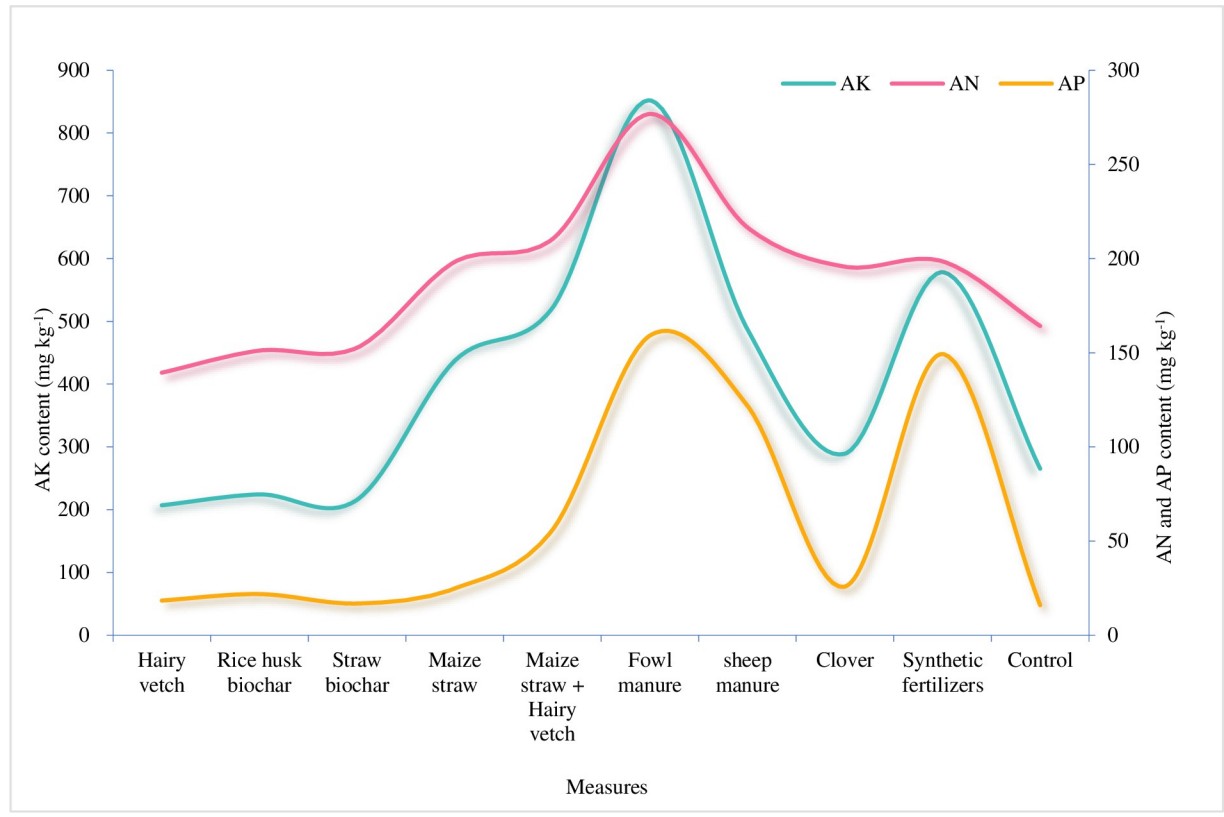

**Fig 5.  Soil available nutrients of soil improve measures.**

Table 4.  Soil nutrient levels under different treatments.

| Measures | Soil nutrient levels | | | | | | | |
|---|---|---|---|---|---|---|---|---|
| | SOM | TN | TP | TK | AN | AP | AK | Sum |
| Fowl manure | 2 | 1 | 2 | 4 | 1 | 1 | 1 | 12 |
| Sheep manure | 2 | 1 | 4 | 4 | 1 | 1 | 1 | 14 |
| Synthetic fertilizers | 3 | 2 | 2 | 4 | 1 | 1 | 1 | 14 |
| Maize straw + Hairy vetch | 2 | 1 | 5 | 4 | 1 | 1 | 1 | 15 |
| Maize straw | 2 | 2 | 4 | 4 | 1 | 2 | 1 | 16 |
| Clover | 2 | 2 | 5 | 4 | 1 | 2 | 1 | 17 |
| Rice husk biochar | 3 | 2 | 5 | 4 | 1 | 2 | 1 | 18 |
| Maize straw biochar | 3 | 2 | 5 | 4 | 1 | 3 | 1 | 19 |
| Control | 3 | 2 | 5 | 4 | 1 | 3 | 1 | 19 |
| Hairy vetch | 4 | 2 | 5 | 4 | 2 | 3 | 1 | 21 |

available nutrient contents in soil (sum of the AN, AP and AK) in descending order were fowl manure (1287.670 mg kg$^{-1}$), synthetic fertilizer (925.889 mg kg$^{-1}$), sheep manure (825.979 mg kg$^{-1}$), maize straw + hairy vetch (787.012 mg kg$^{-1}$), maize straw (660.021 mg kg$^{-1}$), clover (510.899 mg kg$^{-1}$), control group (445.486 mg kg$^{-1}$), rice husk biochar (397.353 mg kg$^{-1}$), maize straw biochar (385.320 mg) kg$^{-1}$), hairy vetch (385.320 mg kg$^{-1}$).

## 3.3 Soil nutrient levels

Soil nutrients are essential to crop growth [44, 45]. Low soil nutrients limit plant growth [46–48], decreasing biomass [49]. Too much soil nutrient content causes environmental and health problems [50] and exerts impacts on terrestrial and aquatic ecosystems [51, 52], resulting in soil degradation and non-point source pollution [53]. Generally, soil fertility can only be evaluated according to nutrient content. As represented in Table 2, the richer the soil nutrient content was, the smaller the number was. Level 1 was the most fertile soil, and level 6 was the most infertile. Adding up the soil nutrient levels of soil organic matter, soil total nutrient and available nutrients, it was found that the smaller sum predicted more fertile soil, and on the contrary, the bigger evinced that the soil was more infertile. As displayed in Table 4, except for hairy vetch and maize straw biochar, the soil fertility under other measures was higher than that of the control, with the best exhibited for fowl manure and the worst displayed for hairy vetch. Among others, the measures of fowl manure, sheep manure, maize straw + hairy vetch, maize straw, and clover significantly enhanced soil organic matter. Only fowl manure and synthetic fertilizer improved soil TP at a higher soil nutrient level. Under all measures, TK lingered at a lower nutrient level, while AN and AK kept the highest. Fowl manure, sheep manure, synthetic fertilizer, maize straw, + hairy vetch significantly improved AP.

## 3.4 Comprehensive evaluation of soil quality

The *SQI* covers the physical and chemical properties of soil and represents the overall quality of soil, which is positively correlated with crop biomass and yield [31, 32]. Fig 6 presents the *SQI* of 9 soil improving treatments and the control group by calculating weight and membership degree. It can be seen that the mean value of the *SQI* was 0.502, and the coefficient of variation was 0.134. The comprehensive evaluation of soil quality by physical and chemical property indexes produced the following findings. The lowest *SQI* was found under maize straw biochar, at the same level as rice husk biochar, both lower than that of the control. Hairy vetch and clover were at the same soil quality level, and so were maize straw and maize straw

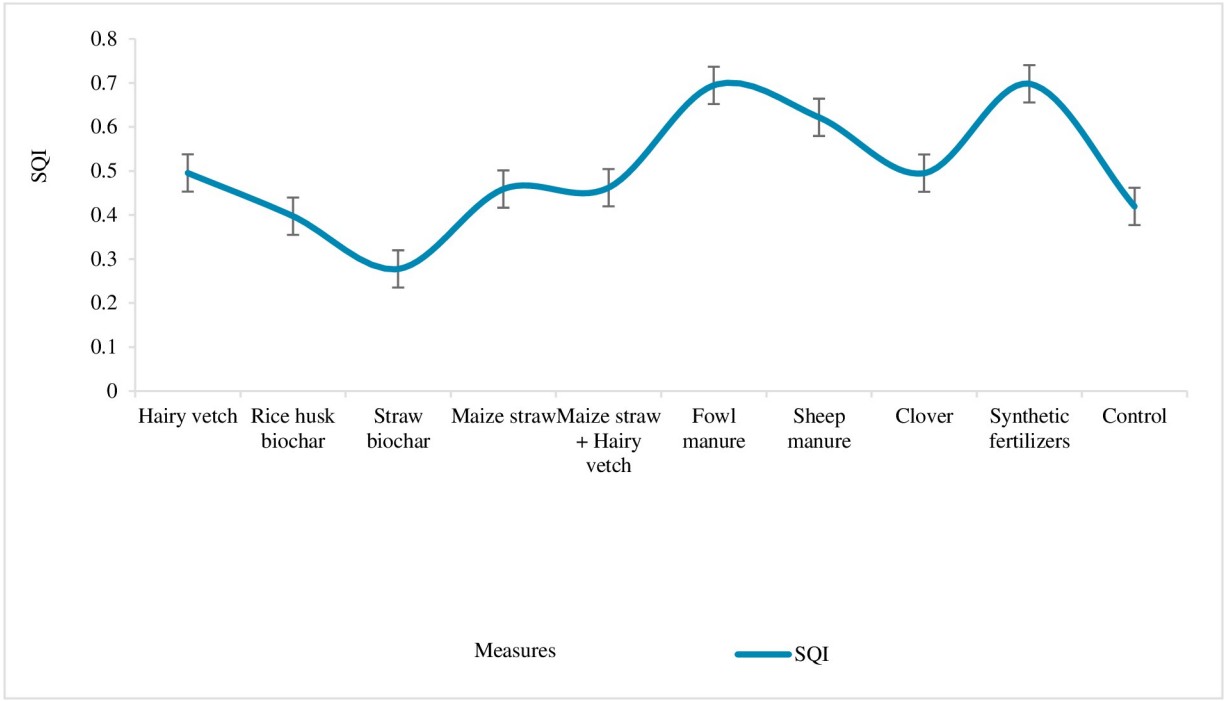

**Fig 6. Comprehensive evaluation of soil quality.**

+hairy vetch. The *SQI* of the two biological measures of hairy vetch and clover was at the same level. The best measures to improve soil quality level were synthetic fertilizer and the two kinds of manure. The *SQI* of the ten plots in descending order were synthetic fertilizer (0.698), fowl manure (0.694), sheep manure (0.622), Hairy vetch (0.4954), clover (0.495), maize straw + hairy vetch (0.462), maize straw (0.459), control group (0.419), rice husk biochar (0.397), maize straw biochar (0.397), and maize straw biochar Carbon (0.278).

## 4 Discussion

### 4.1 Suitability of biological measures for soil improvement

Soil restoration by biological measures has gained widespread trust in the past few decades [54, 55]. The mechanisms behind this treatment are analyzing above-ground and subsurface parts of plants [56, 57]. The above-ground branches and leaves effectively intercept rainfall and reduce soil erosion by runoff and raindrops [58–60]. The deep and shallow roots support the soil's underground parts and lateral roots [61–63], which increases the number of soil aggregates, enhances soil corrosion resistance, and lowers soil erosion, thereby achieving the purpose of improving soil physical structure. Soil remediation measures by using the above-ground and underground parts of plants to reduce soil erosion are generally applied to slope stability management of sloping farmland, which actually utilizes the mechanical and hydro-logical effects of soil improvement [64, 65]. Another advantage of biological measures is to enhance soil fertility and soil nutrients. The Gyeongsang National University Experimental Farm (36˚50′N and 128˚26′E), in Jinju, South Korea, has an average annual temperature of 12.9–13˚C and annual rainfall of 1221–1751 mm. Here, researchers cultivated in rice paddy fields by mixing hairy vetch (*Vicia villosa R.*) with barley (*Hordeum vulgare L.*) and reported that green manure planting improved crop biomass and soil nutrientsv [66]. Other studies

proved this result. It was found in the experiment of planting hairy vetch in Dongxiang County (28°10′N, 106°35′E, average annual precipitation of 1777 mm), Jiangxi Province, in China, that green manure had the function of soil nutrient enhancement. At the Red Soil Experimental Station (26°45′N, 111°52′E; elev. 150 m, annual rainfall of 1422 mm), experiments on planting milk vetch in paddy fields revealed that this kind of plant enhanced rice yield and soil fertility [67–69]. It can be safely concluded that plants, especially green manure, have the effects on decreasing soil erosion, strengthening soil corrosion resistance, and improving crop yield and soil fertility. However, it is worth noting that there was rich precipitation and abundant soil water content in the study areas in the above studies, indicating that sufficient soil water is the necessary medium for the growth of green manure plants to increase soil fertility. In this study, due to high temperatures and little rain during the growth of hairy vetch and clover, the intense evaporation and low soil water content severely hindered the growth of crops. With no ploughing of green manure plants, the biological measures failed to significantly improve the soil. Therefore, under drought conditions, soil improvement with plant measures is highly dependent on artificial water recharge [70], which is time-consuming and laborious with little effect [71].

## 4.2 Suitability of biochar measures for soil improvement

Biochar achieves the purpose of soil restoration by improving soil's physical and chemical properties. In terms of physical properties, biochar promotes soil biological activity, increases soil porosity [72] and water-holding capacity [73], and reduces soil bulk density [74]. About chemical properties, biochar promotes soil organic carbon (SOC) storage in agricultural land [75], brings down soil C and N leaching [76] and nitrate mineralization [77], and enhances soil fertility and crop yield [78]. In the coastal wetland of Jiaozhou Bay, Shandong Province, China, a pot experiment on soil remediation using biochar to restore saline-alkali soil (60% of soil water holding capacity) showed that biochar could contribute to the higher contents of soil SOC, dissolved organic carbon (DOC), TN, TP and AP [79]. In Yifeng Village (31°24 ′10 "N, 119°41′ 28" , annual mean temperature of 17˚C, yearly precipitation of 1242 mm) of Yixing Municipality, Jiangsu Province, China, biochar restoration test on the rice paddy soil showed that the soil carbon content could be increased by 45% with the addition of biochar. In addition, soil agglomeration increased by 30%, and microbial growth and enzyme activity increased similarly [80]. In Nalanda, Bihar, India, a soil remediation experiment (80% of soil water holding capacity) exhibited that biochar could enhance the abilities of cation exchange, anion exchange, and water holding capacity. Improvement was also made in porosity, soil respiration rate, soil contents, total nutrients, and available nutrients [81]. In Owo, Ondo State, Nigeria (average annual rainfall was 1400 mm), biochar improved soil physical properties (reducing soil bulk density, increasing soil porosity and soil water content) and chemical properties (enhancing the content of the SOC, N, P, K and Ca and Mg) to varying degrees, and promoted sweet potato yield as well [82]. Taken altogether, biochar was added for soil restoration under abundant water supply conditions, indicating that sufficient soil water ensures the effective functioning of biochar in soil improvement. In addition, the use of biochar, especially in bulk particle form, is limited in soil remediation [83]. In this study, the biochar derived from maize straw and rice husk was in a large-particle form, significantly reducing their effectiveness for soil remediation. Furthermore, the annual rainfall at the hydropower station site in 2022 was just 850 mm, causing high temperatures and intense evaporation. Hence, the soil remained consistently drought-stressed, impeding biochar's effective functioning in soil improvement.

### 4.3 Problems and countermeasures of fertilizer in soil restoration

Using high dosages of chemical fertilizers, especially nitrogen, enhanced light fraction of soil carbon decomposition [84] and low carbon inputs resulted in a serious decline of the SOM [85], lowering soil fertility and crop yield [86]. The solution to this problem is to provide enough nitrogen to the soil to ensure that the crops can fully absorb it [87, 88] or to increase soil fertility, especially organic matter and carbon content [89–93], or to apply biochar in combination with fertilizer to synchronize soil nutrient availability with crop nitrogen demand [94]. The PMAS-Arid Agricultural University Research Center in Rawalpindi, Pakistan, studied the effects of various biochar and chemical fertilizer combinations against wheat yield and soil quality through pot experiments. The results showed that biochar and chemical fertilizer combinations were more likely to improve cultivated land crop yield and soil quality [86]. A field experiment of combined application of compound fertilizer and farm fertilizer was carried out in farmers' fields in Uttar Chandamari village, Muratipur, Nadia, West Bengal, India (88˚27' E, 22˚59' N). The results were found as: by integrating concentrated organic manures with chemical fertilizer, weed growth and nutrient removal were effectively prevented, significantly enhancing 9% biomass growth, 10% yield of the rice crop along with 3–7% higher nutrient uptake [95]. It can be concluded that combining chemical fertilizer with manure, biochar, and biological fertilizer helps to improve soil fertility more than chemical fertilizer alone. Although the soil quality restoration effect of synthetic fertilizer was the best in this study, chemical fertilizers should be mixed with manure, biochar, and biological fertilizer to avoid the issues caused by long-term use of chemical fertilizer, such as soil compaction, agricultural non-point source pollution, soil fertility decline, crop yield decline [96, 97].

## 5 Conclusions

Soil in the hydropower station construction site was destroyed severely, and restoring it was crucial. There are many measures to improve the soil quality. Implementing nine soil-improving treatments and one control group on the hydropower station construction site helped find the measures that should contribute most to soil improvement.

1. All soil restoring measures benefitted the soil water storage capacity. The soil porosity is significant in the site where the hydropower station was built. Under drought conditions, the soil porosity could lead to the acceleration of soil evaporation and the decrease of soil water content, except for vegetation cover.

2. The SOM content generally becomes higher under soil improvement treatments, and the decomposed organic matter content of maize straw, biochar, fowl manure, and sheep manure increases with time within a year. Manure and straw have advantages in producing more SOM. Synthetic fertilizer, fowl manure, sheep manure, and maize straw have superior effects on increasing the content of soil total nutrients and available nutrients.

3. Manure and synthetic fertilizers exhibit the most significant impact on enhancing soil nutrient levels and overall soil quality. Biological measures rank next, while biochar shows the least effectiveness.

4. The efficacy of biological and biochar measures for soil remediation is constrained by drought conditions, with optimal performance achievable only under abundant rainfall or adequate water supply. Additionally, the particle size of biochar proves to be a significant limiting factor for soil improvement.

5. Prolonged use of high doses of chemical fertilizer, particularly nitrogen fertilizer, can result in soil compaction, non-point source pollution, and a decline in land productivity.

Combining chemical fertilizer with manure, biochar, and biological fertilizer helps mitigate environmental issues associated with sole chemical fertilizer applications and enhances soil fertility, leading to increased crop yield.

## Supporting information

**S1 File. Soil water content.**
(XLSX)

**S2 File. Total soil porosity.**
(XLSX)

**S3 File. Soil organic matter an pH.**
(XLSX)

**S4 File. Soil total nitrogen and available nitrogen.**
(XLSX)

**S5 File. Soil total phosphorus and available phosphorus.**
(XLSX)

**S6 File. Soil total potassium and available potassium.**
(XLSX)

## Acknowledgments

The authors are grateful to the anonymous reviewers for their detailed comments, which have significantly improved the presentation of this work. Readers can also contact the first author via <wuqinglin2007@126.com> for questions about the paper.

## Author Contributions

**Conceptualization:** Qinglin Wu, Panpan Wu.

**Data curation:** Qinglin Wu, Rong Sun, Fan Chen, Xichuan Zhang, Lan Wang, Rui Li.

**Formal analysis:** Qinglin Wu, Lan Wang, Rui Li.

**Funding acquisition:** Qinglin Wu, Rong Sun, Fan Chen, Xichuan Zhang, Rui Li.

**Investigation:** Qinglin Wu, Rong Sun, Fan Chen, Xichuan Zhang, Panpan Wu, Lan Wang, Rui Li.

**Methodology:** Qinglin Wu, Xichuan Zhang, Lan Wang, Rui Li.

**Project administration:** Qinglin Wu, Fan Chen, Xichuan Zhang, Lan Wang, Rui Li.

**Resources:** Qinglin Wu, Rong Sun, Xichuan Zhang, Rui Li.

**Software:** Qinglin Wu, Panpan Wu, Rui Li.

**Supervision:** Qinglin Wu.

**Validation:** Qinglin Wu, Panpan Wu.

**Visualization:** Qinglin Wu, Lan Wang.

**Writing – original draft:** Qinglin Wu, Lan Wang.

**Writing – review & editing:** Qinglin Wu, Lan Wang.

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
