## [Decision Letter · Decision Letter 0]

29 May 2024

PONE-D-24-07846Soil quality enhancement by multi-treatment in the abandoned land of dry-hot river valley hydropower station construction area under karst desertification environmentPLOS ONE

Dear Dr. Li,

Thank you for submitting your manuscript to PLOS ONE. After careful consideration, we feel that it has merit but does not fully meet PLOS ONE’s publication criteria as it currently stands. Therefore, we invite you to submit a revised version of the manuscript that addresses the points raised during the review process.

We look forward to receiving your revised manuscript.

Kind regards,

Paulo H. Pagliari

Academic Editor

PLOS ONE

Journal Requirements:

"This study was supported by the Technology Support Plan of Guizhou Province (No. 462 2021 Qiankehe Zhicheng) and the Natural Science Foundation of Guizhou Province (No. 317 2022 Qiankehe Jichu -ZK)."

"This study was supported by the Technology Support Plan of Guizhou Province (No. 462 2021 Qiankehe Zhicheng) and the Natural Science Foundation of Guizhou Province (No. 317 2022 Qiankehe Jichu -ZK)."

"The author(s) received no specific funding for this work"

6. We note that your Data Availability Statement is currently as follows: All relevant data are within the manuscript and its Supporting Information files

7. PLOS requires an ORCID iD for the corresponding author in Editorial Manager on papers submitted after December 6th, 2016. Please ensure that you have an ORCID iD and that it is validated in Editorial Manager. To do this, go to ‘Update my Information’ (in the upper left-hand corner of the main menu), and click on the Fetch/Validate link next to the ORCID field. This will take you to the ORCID site and allow you to create a new iD or authenticate a pre-existing iD in Editorial Manager. Please see the following video for instructions on linking an ORCID iD to your Editorial Manager account: https://www.youtube.com/watch?v=_xcclfuvtxQ

Reviewers' comments:

Reviewer's Responses to Questions

**Comments to the Author**

1. Is the manuscript technically sound, and do the data support the conclusions?

Reviewer #1: Yes

2. Has the statistical analysis been performed appropriately and rigorously? 

Reviewer #1: Yes

3. Have the authors made all data underlying the findings in their manuscript fully available?

Reviewer #1: Yes

4. Is the manuscript presented in an intelligible fashion and written in standard English?

Reviewer #1: Yes

5. Review Comments to the Author

Reviewer #1: Conducted in the dry-hot riever valley hydropower station construction area in karst desertification environment, this study adopted multiple measures to improve soil physical and chemical properties of soil, and comprehensively evaluated soil quality through soil fertility criteria and principal component analysis (PCA), with abundant data and good research results. The controversial research results are discussed in depth, and finally a convincing conclusion is drawn. This study makes contribution to soil improvement under special environment through taking a variety of soil improvement measures and carry out comprehensive evaluation. The research results provide an certain reference for the treatment and improvement of soil ecosystem. Yet, some minor problems still exist. It would be better if the following ones could be solved.

1.In the part of abstract, it is suggested to add some data to the research results to enhance the persuasiveness.

2.Line 23: The “different measures” should be replaced with “multi-treatment”.

3.Key words should include “construction area”, which is an important research object that distinguishes this article from other non-construction site articles.

4.Line 58-88: The presentation of scientific questions should also highlight the rarity of comprehensive evaluation of soil fertility and soil quality using multiple soil improvement measures.

5.The conclusion is suggested being written in items.

6.Use English references whenever possible.

6. PLOS authors have the option to publish the peer review history of their article (what does this mean?). If published, this will include your full peer review and any attached files.

Reviewer #1: No

---

## [Author Response · Author response to Decision Letter 0]

10 Jun 2024

Response to reviewers

Reviewer #1: Conducted in the dry-hot riever valley hydropower station construction area in karst desertification environment, this study. adopted multiple measures to improve soil physical and chemical properties of soil, and comprehensively evaluated soil quality through soil fertility criteria and principal component analysis (PCA), with abundant data and good research results. The controversial research results are discussed in depth, and finally a convincing conclusion is drawn. This study makes contribution to soil improvement under special environment through taking a variety of soil improvement measures and carry out comprehensive evaluation. The research results provide an certain reference for the treatment and improvement of soil ecosystem. Yet, some minor problems still exist. It would be better if the following ones could be solved.

1.In the part of abstract, it is suggested to add some data to the research results to enhance the persuasiveness.

Response: We have added necessary data in the summary to improve the credibility of our findings.

2.Line 23: The "different measures" should be replaced with "multi-treatment".

Response: We have replaced "different measures" with "multi-treatment" in the revised version.

3.Key words should include "construction area", which is an important research object that distinguishes this article from other non-construction site articles.

Response: The construction area has been added to the keyword section.

4.Line 58-88: The presentation of scientific questions should also highlight the rarity of comprehensive evaluation of soil fertility and soil quality using multiple soil improvement measures.

Response: In the last paragraph of introduction section, we’ve added the summary of the shortcomings of previous studies on soil improvement, laying the groundwork for proposing the breakthrough point of this study.

5.The conclusion is suggested being written in items.

Response: The conclusion has been listed in five items.

(1)All soil restoring measures benefitted the soil water storage capacity. The soil porosity is significant in the site where the hydropower station was built. Under drought conditions, the soil porosity could lead to the acceleration of soil evaporation and the decrease of soil water content, except for vegetation cover. 

(2) The SOM content generally becomes higher under soil improvement treatments, and the decomposed organic matter content of maize straw, biochar, fowl manure, and sheep manure increases with time within a year. Manure and straw have advantages in producing more SOM. Synthetic fertilizer, fowl manure, sheep manure, and maize straw have superior effects on increasing the content of soil total nutrients and available nutrients. 

(3)Manure and synthetic fertilizers exhibit the most significant impact on enhancing soil nutrient levels and overall soil quality. Biological measures rank next, while biochar shows the least effectiveness. 

(4)The efficacy of biological and biochar measures for soil remediation is constrained by drought conditions, with optimal performance achievable only under abundant rainfall or adequate water supply. Additionally, the particle size of biochar proves to be a significant limiting factor for soil improvement. 

(5)Prolonged use of high doses of chemical fertilizer, particularly nitrogen fertilizer, can result in soil compaction, non-point source pollution, and a decline in land productivity. Combining chemical fertilizer with manure, biochar, and biological fertilizer helps mitigate environmental issues associated with sole chemical fertilizer applications and enhances soil fertility, leading to increased crop yield. 

6.Use English references whenever possible.

Response: After careful examination, we found that there were 8 Chinese references in the original manuscript (10,22,34,35,40,42,67,95, respectively), among which we could only find 4 relevant and supporting English references to replace 10,22,42,95.

---

## [Decision Letter · Decision Letter 1]

16 Jun 2024

Soil quality enhancement by multi-treatment in the abandoned land of dry-hot river valley hydropower station construction area under karst desertification environment

PONE-D-24-07846R1

Dear Dr. Li,

We’re pleased to inform you that your manuscript has been judged scientifically suitable for publication and will be formally accepted for publication once it meets all outstanding technical requirements.

Kind regards,

Paulo H. Pagliari

Academic Editor

PLOS ONE

Additional Editor Comments (optional):

We can now accept your manuscript for publication. Congratulations!

Reviewers' comments:

Reviewer's Responses to Questions

**Comments to the Author**

1. If the authors have adequately addressed your comments raised in a previous round of review and you feel that this manuscript is now acceptable for publication, you may indicate that here to bypass the “Comments to the Author” section, enter your conflict of interest statement in the “Confidential to Editor” section, and submit your "Accept" recommendation.

Reviewer #1: All comments have been addressed

2. Is the manuscript technically sound, and do the data support the conclusions?

Reviewer #1: Yes

3. Has the statistical analysis been performed appropriately and rigorously? 

Reviewer #1: Yes

4. Have the authors made all data underlying the findings in their manuscript fully available?

Reviewer #1: Yes

5. Is the manuscript presented in an intelligible fashion and written in standard English?

Reviewer #1: Yes

6. Review Comments to the Author

Reviewer #1: You have provided very interesting work. Thank you for submitting such a carefully revised manuscript and rich raw data. Congratulations.

7. PLOS authors have the option to publish the peer review history of their article (what does this mean?). If published, this will include your full peer review and any attached files.

Reviewer #1: No

---

## [Editor Report · Acceptance letter]

24 Jun 2024

PONE-D-24-07846R1 

PLOS ONE

Dear Dr. Li, 

I'm pleased to inform you that your manuscript has been deemed suitable for publication in PLOS ONE. Congratulations! Your manuscript is now being handed over to our production team.

Kind regards, 

on behalf of

Dr. Paulo H. Pagliari 

Academic Editor

PLOS ONE